# Acute Kidney Injury in COVID-19

**DOI:** 10.3390/ijms22158081

**Published:** 2021-07-28

**Authors:** Marta Głowacka, Sara Lipka, Ewelina Młynarska, Beata Franczyk, Jacek Rysz

**Affiliations:** Department of Nephrology, Hypertension and Family Medicine, Medical University of Lodz, 90-549 Lodz, Poland; marta.glowacka96@gmail.com (M.G.); saralipka@wp.pl (S.L.); bfranczyk-skora@wp.pl (B.F.); jacek.rysz@umed.lodz.pl (J.R.)

**Keywords:** COVID-19, acute kidney injury, RRT, dialysis

## Abstract

COVID-19 is mainly considered a respiratory illness, but since SARS-CoV-2 uses the angiotensin converting enzyme 2 receptor (ACE2) to enter human cells, the kidney is also a target of the viral infection. Acute kidney injury (AKI) is the most alarming condition in COVID-19 patients. Recent studies have confirmed the direct entry of SARS-CoV-2 into the renal cells, namely podocytes and proximal tubular cells, but this is not the only pathomechanism of kidney damage. Hypovolemia, cytokine storm and collapsing glomerulopathy also play an important role. An increasing number of papers suggest a strong association between AKI development and higher mortality in COVID-19 patients, hence our interest in the matter. Although knowledge about the role of kidneys in SARS-CoV-2 infection is changing dynamically and is yet to be fully investigated, we present an insight into the possible pathomechanisms of AKI in COVID-19, its clinical features, risk factors, impact on hospitalization and possible ways for its management via renal replacement therapy.

## 1. Introduction

COVID-19 is an infectious disease caused by severe acute respiratory syndrome coronavirus 2 (SARS-CoV-2), and it originated in Wuhan, China, in December 2019. As of 4 March 2021, approximately 84,230,049 cases have been discovered worldwide, causing an ongoing global pandemic [1]. Such a large number of confirmed cases is related to the way the virus is transmitted, which is close human-to-human contact through droplets or aerosol via coughs, sneezes or talking. Infection may also occur by touching contaminated surfaces and then touching routes of transmission such as the mouth, eyes or nose [2]. COVID-19 disease mainly affects the respiratory system, which in more severe cases is manifested by pneumonia, hypoxemia and acute respiratory distress syndrome. Although the main focus is on the pulmonary features, physicians must be aware of complications that SARS-CoV-2 infection carries to other organs, including the kidneys [3]. Acute kidney injury (AKI) is the most common kidney manifestation among patients hospitalized with COVID-19. According to KDIGO, AKI is defined as any of the following: (1) an increase in serum creatinine (SCr) by ≥0.3 mg/dL (≥26.5 μmol/L) within 48 h; or (2) an increase in SCr ≥ 1.5 times of baseline within the prior 7 days; or (3) urine volume < 0.5 mL/kg/hour for 6 h. AKI can be also staged for severity according to KDIGO: stage (1) increase in SCr to 1.5–1.9 times baseline or by ≥0.3 mg/dL; stage (2) increase SCr to 2.0–2.9 times baseline; stage (3) increase SCr to 3.0 times baseline or increase in serum creatinine to ≥4.0 mg/dL (≥353.6 mmol/L) or initiation of renal replacement therapy [4]. Moreover, AKI development is far more frequent in severe and critically ill patients and is associated with poor prognosis and higher mortality [5,6,7]. Therefore, understanding the underlying pathophysiology of kidney injury in the course of COVID-19 is crucial for early recognition of the damage and the implementation of optimal treatment.

## 2. Epidemiology

With 84,230,049 cases of SARS-CoV-2 infection globally, at least one third is asymptomatic [8]. Among those COVID-19 patients who experience symptoms, about 80% develop mild to moderate symptoms, while 20% of cases present with severe symptoms over the course of the disease, of whom 6% become critically ill [9]. The overall mortality rate of COVID-19 patients is around 3% but, in the critically ill group, it can reach 50% [10].

The incidence of acute kidney injury in COVID-19 varies in different case reports. Studies in China [5,10,11] have shown that AKI occurred in 5% to 29% within a median of 7–14 days after admission, whereas reports from the United States [6,12,13] have shown greater rates reaching from 37% to 57% in COVID-19 positive patients. However, the onset of AKI in the US was observed much earlier—either upon admission or within 24 h of admission. Another study from Brazil also showed a high occurrence of AKI in 56% of COVID-19 patients, of which 67% developed stage 3 AKI [14].

Fisher et al. [13] presented a comparison report between 3345 patients with COVID-19 and 1265 patients without COVID-19 during the same hospitalization period. AKI development in the COVID-19 (+) group was higher than in the COVID-19 (−) group (57% and 37%, respectively), and a significant number of patients positive for COVID-19 had stage 3 AKI compared with patients negative for COVID-19 (17.2% vs. 7.3%). Moreover, 4.9% of the patients positive for COVID-19 required renal replacement therapy (RRT) compared with 1.6% of those negative for COVID-19. Other US studies reported RRT necessity in up to 19% of patients with COVID-19, while in Brazil it was up to 47% of patients.

The results of incidence of AKI in COVID-19 patients are presented in Table 1.

## 3. Mechanism of SARS-CoV-2 Cellular Kidney Infection

It is now a well-known fact that the main target of SARS-CoV-2 is the lungs; more precisely, type II pneumocytes. More and more studies published to date have proved that not only the lungs are exposed to infection, but so are the heart, liver, gastrointestinal tract, bone marrow and kidneys [3,16] Multiorgan tropism is due to the fact that SARS-CoV-2 gains access to the cells through an endogenous viral receptor angiotensin converting enzyme 2 (ACE2) [17].

In order for SARS-CoV-2 to enter the host, cells are required to bind its transmembrane spike (S) glycoprotein to cellular receptor ACE2. S consists of two subunits, each with a different function. S1 is responsible for binding to the host cell receptor, while S2 is used to fuse the viral membrane with the membrane of the infected cell [18,19]. Spike then requires proteolytic priming to be activated, which is granted by serine protease TMPRSS2. Therefore, ACE2 and TMPRSS co-expression is a key determinant for the entry of SARS-CoV-2 into host cells [20,21]. Once SARS-CoV-2 is in the cytosol of the infected cell, the translation of its RNA and virion synthesis begins. It has been proven that genomic replication and virion assembly occur within the double vesicles of the endoplasmic reticulum (ER) and the Golgi complex [22].

We can therefore conclude that susceptible kidney cells are those that express ACE2. Using RNA-Seq sequencing techniques, scientists were able to determine which kidney cell types comprised the ACE2 gene. The data show that ACE2 mRNA is mostly expressed in proximal tubular epithelial cells and podocytes [20,21]. This would concur with a report by Braun F. et al., who managed to isolate SARS-CoV-2 from epithelial cells of an autopsied kidney [23].

## 4. Pathophysiology

The best understood mechanism of kidney damage induced by SARS-CoV-2 is direct cellular infection. However, there are also few possible reasons for acute renal failure, such as inflammatory damage caused by cytokine storm, AKI related to acute respiratory distress syndrome (ARDS), kidney–lung crosstalk theory, hypovolemia and collapsing glomerulopathy [24,25,26].

### 4.1. Direct Cellular Infection

As mentioned before, SARS-CoV-2 penetrates through angiotensin converting enzyme 2. The highest concentration of ACE2 in the kidneys was proven to be located in proximal tubular epithelial cells and podocytes [20,21,27]. Therefore, the direct infection of kidney cells by SARS-CoV-2 virus is the most likely mechanism for the development of acute kidney injury. Autopsy data also speak to this mechanism of AKI because many researchers found virus-like particles in epithelial cells of kidneys [24].

### 4.2. Cytokine Storm and AKI Related to ARDS

The abnormal immune response associated with SARS-CoV-2 is also a likely mechanism for the development of acute renal failure. At the root of these irregularities lies the cytokine storm and leukopenia [24]. Sepsis, a hemophagocytic syndrome, can lead to a so-called “cytokine storm”, which is a cytokine release syndrome (CRS). The most crucial cytokine responsible for this pathology is IL-6 [25]. IL-6 also occurs in ARDS complications of COVID-19. AKI in CRS is on intrarenal inflammation, increased vascular permeability, volume depletion and cardiomyopathy grounds. Cardiomyopathy can cause stasis in the renal veins, resulting in renal hypotension and hypoperfusion, leading to a reduction in the glomerular filtration rate. This phenomenon is called the syndrome type 1, which is manifested by endothelial damage, pleural effusions, edema, intra-abdominal hypertension, third-space fluid loss, intravascular fluid depletion and hypotension [28]. Moreover, other complications of COVID-19, such as right and left ventricular failure, can cause AKI. The first one leads to blood stagnation in the kidneys, whereas the second one to reduced cardiac output and then to renal hypoperfusion [24]. There are five causes of AKI in ARDS: hemodynamic instability, hypoxemia/hypercapnia, acid-base dysregulation, inflammation and neurohormonal effects [25].

### 4.3. Lung–Kidney Crosstalk Theory

The occurrence of kidney dysfunctions in COVID-19 patients might be explained by the kidney–lung crosstalk theory. This is due to the increased concentration of cytokines in the blood, the release of which is promoted by lung injury. Elevated levels of cytokines, especially IL-6, cause an increase in alveolar capillary permeability and pulmonary hemorrhage, and may even lead to distant-end organ dysfunction as damage to vascular endothelium in the kidneys. As a consequence, it leads to secondary hypoxia of the kidney and further damage to its structures. In patients who did not have chronic kidney disease or AKI, the research has shown that most of them manifested ARDS and/or AKI after developing pneumonia, which also testifies to the presence of lung–kidney crosstalk [28].

### 4.4. Hypovolemia

An incorrect distribution of fluids, especially hypovolemia (a consequence of fever and tachypnea), may affect the kidneys. This condition causes renal hypoperfusion and, consequently, renal failure. Endothelial damage, loss of fluid into the third space and hypotension provoke renal hypoperfusion. Virus cells and cytokines produced by the organism destroy the endothelium, which causes edema, ascites and hydrothorax, which in turn leads to hypotension and a loss of fluids to the third space. The amount of circulating fluid is reduced and this damages the kidneys in the prerenal mechanism. [28] Many patients infected with SARS-CoV2 have gastrointestinal symptoms that greatly increase the loss of fluid and further dehydration of the patient, mainly leading to pre-renal AKI [25]. In hemodynamically unstable patients, the venous flow deteriorates. Rhabdomyolysis (a condition involving the rapid dissolution of damaged or injured skeletal muscle), metabolic acidosis (increased level of hydrogen ions in the systemic circulation, which results in a reduction in the level of serum bicarbonate) and hyperkalemia (serum or plasma potassium level greater than 5.0 mEq/L to 5.5 mEq/L) are also associated with this condition. This has a significant impact on the degradation of kidney function and, later on, on the occurrence of AKI. [28] Therefore, optimizing hemodynamic is crucial to kidneys health.

### 4.5. Collapsing Glomerulopathy

Collapsing glomerulopathy (CG) is a histological term for focal segmental glomerulosclerosis defined by segmental or global glomerular collapse correlated with podocyte proliferation, whose typical feature is proteinuria. CG has been associated with various factors, but the essential one is the presence of Apolipoprotein 1 (APOL1) genotype, namely alleles G1 and G2 [29]. Since the COVID-19 outbreak, there have been several case reports presenting COVID-19 patients with CG, in which authors speculated about the possible mechanisms linking it to SARS-CoV-2 infection. Kissling et al. [30] suggested direct viral toxicity on tubular cells as a pathomechanism of acute tubular necrosis in COVID-19 patients with G1 variant of APOL1 gene, based on post-mortem kidney examination. In another report, Peleg et al. [31] failed to detect viral particles in kidney tissue; thus, cytokine storm was presumed as a cause of collapsing glomerulopathy in COVID-19 patients. Both theories are possible, especially with high-risk alleles of the APOL1 gene.

## 5. Histopathology

Many autopsies have been carried out since the beginning of the pandemic on COVID-19 patients, especially in the search of complications the SARS-CoV-2 virus carries to the human cells. Histopathological examinations used, among others, light microscopy, electron microscopy and immunofluorescence.

COVID-19 patients with AKI rarely showed clinical symptoms. The kidneys were often atypical in macroscopic examination, sometimes enlarged. However, microscopic examination of the subjects showed acute tubular injury (ATI), tubulointerstitial injury and glomerular injury, but nevertheless mild in relation to the degree of AKI and blood creatinine concentration [15,32]. Other notable observations were: fibrosis, congestion of the glomeruli and periurethral capillaries and the presence of glomerular fibrin. In addition, changes in the kidneys included atherosclerosis, foci of ischemia, benign chronic glomerular and tubulointerstitial lesions. Moreover, microscopy revealed the presence of isometric vacuolization in the renal tubules as an important diagnostic point. These changes in COVID-19 patients confirm direct viral infection into cells as a mechanism of AKI. Moreover, they correlate with the presence of double-membrane-covered vesicles that contain vacuoles and may be an indicator of active SARS-CoV-2 infection [32,33].

Acute tubular injury (ATI) is the constant condition found in COVID-19 patients with AKI. ATI is characterized by the loss of the brush rim, degeneration of the vacuole, flattening of the lumen of the tubule, cellular inclusions as well as necrosis and detachment of the epithelium. These changes can be well observed in samples subjected to light microscopy. The presence of hemosiderin and fibrin in the renal tubules has been rarely observed [15,23,32]. However, no aggregated platelets were observed at all. Electron microscopy revealed corona-like virus-like cells in the proximal part of the renal tubules and in podocytes. Erythrocytes were present in the lumen of the periurethral vessels [23]. The footprint of interferon in capillaries in electron microscopy indicates a cytokine storm as a mechanism for kidney damage. The visible swelling of the endothelial cells is also evidence of damage to the endothelium [15]. The most frequent histological observations are presented in Table 2.

## 6. Clinical Features

As mentioned before, COVID-19 can be manifested as a mild or moderate infection or a severe or critical illness. Mild or moderate symptoms include cough, fever, fatigue, dyspnea and smell and taste loss. Severe COVID-19 cases additionally present with hypoxia and >50% lung infiltration on imaging. The critical course of the disease includes respiratory failure, SIRS and/or multiple organ failure [9].

Although the main feature of COVID-19 is pneumonia, we focused our research on renal dysfunction. The most frequently reported disorder of the kidney was acute kidney injury (AKI). In researched studies [5,6,11,12,13,16], the end point of AKI was defined by the “Kidney Disease: Improving Global Outcomes (KDIGO)” criteria explained above. These findings indicate that increase in serum creatinine in COVID-19 patients at admission can be a negative prognostic factor in AKI development. In addition, majority of patients who developed AKI presented with hematuria and proteinuria, although these were more frequent in severe or critically ill patients (Table 3).

More research is needed to determine whether the occurrence of AKI affects subsequent renal function. A study with a follow-up period of 3 weeks from the onset of the infection presented no improvement in kidney functions in 89.5% of COVID-19 patients who developed AKI [5]. Another study shows that 46% of COVID-19 patients who had AKI at discharge did not recover to baseline serum creatinine levels [12]. Therefore, the assumption that AKI may lead to CKD is possible but needs further investigation.

## 7. Risk Factors of AKI Development

Most of the patients who experience the severe or critical course of COVID-19 have pre-existing conditions. The most common comorbidities are hypertension and other cardiovascular disorders, diabetes mellitus and obesity. They are considered to be the major risk factors for developing a more severe, if not critical, course of COVID-19 [35]. A meta-analysis by Henry and Lippi pointed out that patients with chronic kidney failure must also take precautions in exposure to SARS-CoV-2 virus, since CKD increases the risk of a severe course of the disease [36].

In our research, we focused on the variables that specifically led to AKI occurrence in COVID-19 patients. Hirsch et al. analyzed possible risk factors associated with AKI development which included older age, diabetes, hypertension, cardiovascular disease and respiratory failure, the last one being the most important factor [6]. In another study, there was an association between older age and male sex as a primary risk factors of developing AKI [13]. Thus, not only do comorbidities play a role in the incidence of AKI, but so do non-modifiable factors such as age and gender. The results of variable correlations with AKI development are presented in Table 4.

Furthermore, the issue of angiotensin-converting enzyme inhibitors (ACEI) and angiotensin-receptor blockers (ARB) in patients with COVID-19 infection and putative risk of AKI in COVID19 is worth addressing. The clinical use of ACEI and ARB was controversial at the beginning of the pandemic. Some researchers proposed, based on the mechanism of SARS-CoV-2 infection, that administering ACEI and ARB could aggravate the course of the disease. This speculation has been dismissed by several observational studies, as well as international societies such as The Council on Hypertension of the European Society of Cardiology. ACEI and ARB therapy should not be discontinued in the mild and/or moderate course of COVID-19 infection [37,38].

## 8. Impact of AKI Development on Hospitalization and Mortality Rate

In order to assess the impact of AKI development in patients with COVID-19 on the course of hospitalization and mortality rate, we have compiled the data from five different case reports and compared AKI patients with patients without AKI (Figure 1, Figure 2 and Figure 3). The occurrence of AKI in COVID-19-positive patients resulted in a significantly increased number of admissions to the intensive care unit (ICU) in comparison to COVID-19 patients without AKI. In the same group of patients, the need for mechanical ventilation was also more noticeable. What stands out the most in our sheet is the fatality rate, which ranges from 33.3% up to 86.4% in COVID-19 patients with AKI in comparison to COVID-19 patients without AKI, which varies from 5.6% to 9.3%. Therefore, patients with COVID-19 who develop AKI are at significantly higher risk of severe or critical course of the disease, respiratory failure and consequent mechanical ventilation. Moreover, AKI incidence undeniably increases the mortality rate in COVID-19 patients.

## 9. Clinical Handling

There is no specific treatment for AKI caused by COVID-19, and it has to be based on KDIGO and other guidelines. Prevention of AKI mainly consists of individual fluid therapies (balanced crystalloids), discontinuation or reduction in nephrotoxic drugs and, in the case of hypovolemia, the use of vasopressors. It is important to monitor kidney function with laboratory tests such as serum creatinine and urea levels.

Treatment should be systemic. We use antiviral drugs, antibiotics, corticosteroids, renin–angiotensin inhibitors, statins and anticoagulants. Drug therapy reduces the risk of developing AKI and causes a milder disease course [39].

## 10. Renal Replacement Therapy

Renal replacement therapy (RRT) is needed in up to 64% of critically ill COVID-19 patients who develop AKI (Table 1) [6,12,13,14,15]. It is due to abnormal concentrations of electrolytes and volume overload resistance to pharmacological treatment. The earliest studies suggested that, in hemodynamically unstable patients with COVID-19, the recommended approach was continuous RRT (CRRT) [28]. However, clinicians observed a significantly increased incidence of circuit clotting in COVID-19 patients, leading to prolonged time of treatment, as well as the unnecessary use of resources [24,40]. Therefore, we must take caution in choosing the most safe and effective strategy.

Ramirez-Sandoval, J.C. et al. [41] reported the viability and safety of prolonged intermittent renal replacement therapy (PIRRT) as a treatment option in COVID-19 patients with AKI. The use of low-molecular-weight heparin (LMWH) in systemic anticoagulation and unfractionated heparin (UFH) in regional anticoagulation was adapted and shown to reduce the incidence of the circuit clotting observed in up to 13% of ICU patients with COVID-19 infection. These findings were later confirmed in a report by Di Mario, F. et al. [42], who tested the effectiveness of sustained low-efficiency dialysis (SLED), which is a modality of PIRRT. However, in this study, the applied anticoagulation was regional citrate anticoagulation (RCA) protocol, and it significantly lowered dialysis interruption by circuit clotting to 6.1%. Moreover, the advantages of RCA over systemic heparin anticoagulation protocols have been demonstrated in a comparative study by Arnold, F. et al. [43] Taking into account these reports, we can assume that SLED with RCA protocol is the safest and most efficient way of AKI management in hemodynamically unstable COVID-19 patients.

## 11. Conclusions

Acute kidney injury is common amongst patients with SARS-CoV-2 infection, especially critically ill ones, and is without a doubt associated with higher mortality. There are numerous possible pathomechanisms which are still being investigated, but the most probable ones are direct cellular invasion, ARDS, cytokine storm and hypovolemia. Histopathological reports showed that most COVID-19 patients with AKI presented acute tubular damage, sometimes with necrosis and collapsing glomerulopathy. The most important steps that should be taken in AKI prevention are the following: minimizing the risk of hypovolemia and monitoring serum creatinine levels in the early stages of COVID-19 infection, especially in regard to the high-risk patients at an older age with diabetes mellitus, hypertension and cardiovascular diseases. However, when, despite our best efforts, COVID-19 patients are hemodynamically unstable, the safest and most efficient way of AKI management is SLED with RCA protocol. The evidence related to COVID-19 is changing dynamically and we are still in need of more research for establishing the optimal treatment of AKI in COVID-19 infection. Furthermore, physicians must be aware that patients who recover from AKI induced by SARS-CoV-2 require monitoring of their kidneys on follow-up, as there is rising evidence showing eGFR decreases among patients with a history of COVID-19-associated AKI [44].

## Figures and Tables

**Figure 1 ijms-22-08081-f001:**
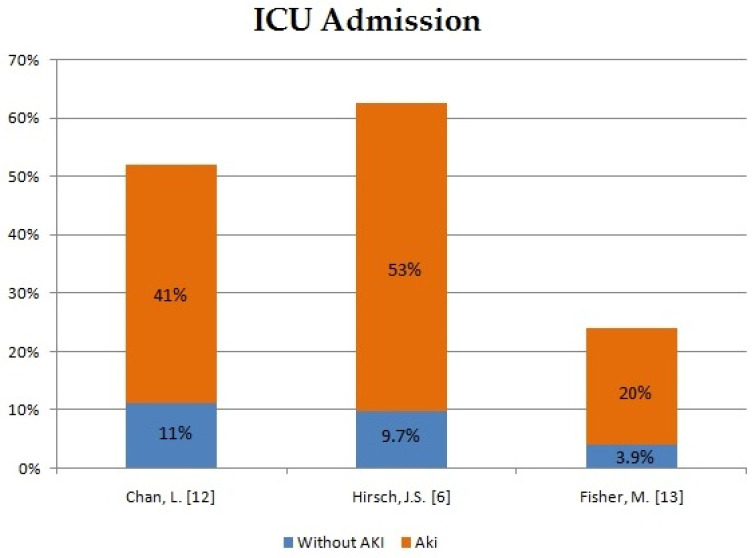
Intensive care unit admission in patients with AKI and no AKI [6,12,13].

**Figure 2 ijms-22-08081-f002:**
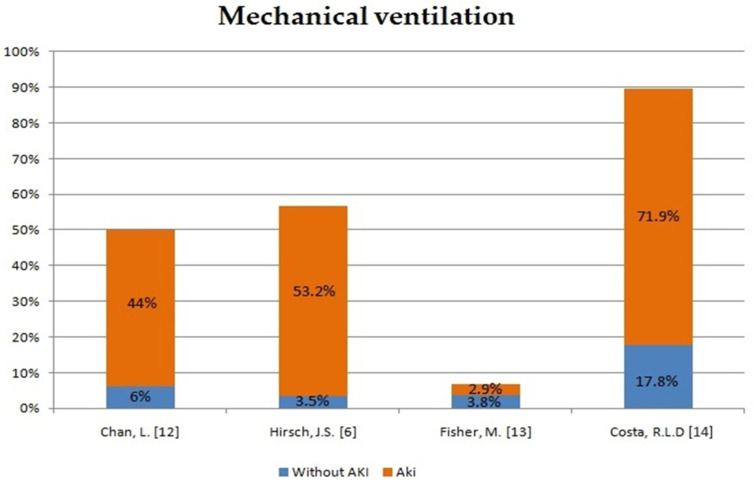
The need for mechanical ventilation in patients with AKI and no AKI [6,12,13,14].

**Figure 3 ijms-22-08081-f003:**
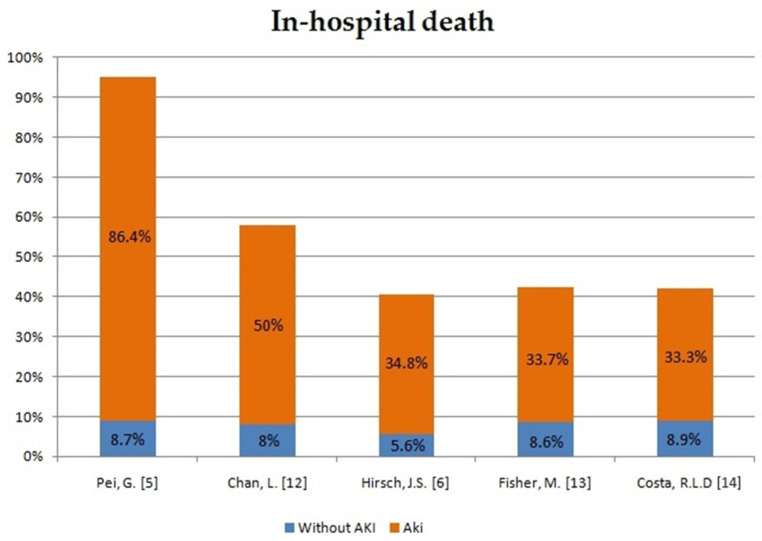
In-hospital death in patients with AKI and no AKI [5,6,12,13,14].

**Table 1 ijms-22-08081-t001:** Incidence of AKI in COVID-19 patients.

	All Patients	AKI	Stages of AKI			RRT in Patients with AKI
			AKI 1	AKI 2	AKI 3	
Yang, X. [10]	52	15 (29%)				
Pei, G. [5]	333	22 (6.6%)	4 (18.2%)	7 (31.8%)	11 (50%)	
Cheng, Y. [11]	701	36 (5.1%)	13 (1.9%)	9 (1.3%)	14 (2%)	
Chan, L. [12]	3993	1835 (45.9%)	387 (21%)	199 (10.8%)	491 (26.7%)	347 (19%)
Hirsch, J.S. [6]	5449	1993 (36.6%)	927 (46.5%)	447 (22.4%)	619 (31.1%)	14.3%
Fisher, M. [13]	3345	1903 (56.9%)	942 (49.5%)	387 (20.3%)	574 (30.2%)	164 (4.9%)
Costa, R.L.D. [14]	102	57 (55.9%)	10 (17.5%)	9 (15.8%)	38 (66.7%)	27 (47.4%)
Ferlicot, S. [15]	47	1 (2.2%)	3 (6.4%)	2 (4.3%)	41 (87.2%)	30 (63.8%)

**Table 2 ijms-22-08081-t002:** The most common histopathological observations in COVID-19.

Morphological Data	Nephron Segments	Pathophysiological Mechanism
Epithelial necrosis	Proximal tubules	Direct viral infections, hemodynamic disorders, rhabdomyolysis
Cellular debris	Lumen of tubules	
Myoglobin	Tubules	Rhabdomyolysis
Corona-like viruses	Podocytes, tubules	Direct viral infections
Isometric vacuolization	Tubules	
Loss of brush border, flattening, damage	Tubules	Direct viral infections, cytokine storm
Swelling of endothelial cells	Glomerulus	

**Table 3 ijms-22-08081-t003:** Laboratory data in AKI patients with COVID-19.

	Pei, G. [5] *n* = 35	Cheng, Y. [11] *n* = 53	Chan, L. [12] *n* = 656	Hirsch, J.S. [6] *n* = 1993	Li, Z. [34] *n* = 147
Proteinuria			84%	26.0%	88/147 (60%)
negative	4/35 (11.4%)	16/53 (30.2%)		168 (26.0%)	31 (21%)
+	24/35 (68.6%)	21/53 (39.6%)		206 (31.9%)	39 (27%)
++/+++	7/35 (20.0%)	16/53 (30.2%)		194 (30.0%)	15 (10%)
+++				78 (12.1%)	3 (2%)
*p*	0.002	<0.001			
Hematuria			81%	46.1%	71/147 (48%)
negative	14/35 (40.0%)	25/53 (47.2%)		196 (36.2%)	21 (14%)
+	13/35 (37.1%)	16/53 (30.2%)		96 (17.7%)	21 (14%)
++/+++	8/35 (22.9%)	12/53 (22.6%)		148 (27.3%)	16 (11%)
+++				102 (18.8%)	13 (9%)
*p*	0.007	<0.001			
Serum creatinine					
Serum creatinine, mg/dL	0.79 (0.64–0.95)	1.49 ± 0.44	1.42 (0.95–2.25)	1.23 (0.91, 1.8)	0.75 (0.61–0.93)
Peak serum creatinine, mg/dL		1.84 ± 1.23		2.23 (1.40, 4.12)	
Blood urea nitrogen					
Blood urea nitrogen, mg/dL	12.04 (9.0–16.0)	30.8 ± 19.6	31 (18–51)	23.0 (14.75, 37.0)	0.01

+—slight; ++—average; +++—significant.

**Table 4 ijms-22-08081-t004:** The correlation between age, sex, comorbid conditions and AKI development.

	Pei, G. [5] *n* = 19	Chan, L. [12] *n* = 1835	Hirsch, J.S. [6] *n* = 1993	Fisher, M. [13] *n* = 1903
Age (years)	64.0 ± 8.1	71 (61–81)	69.0 (58.0, 78.0)	67.1 (15.3)
Sex (%)				
Men	14 (73.7)		1270 (63.7)	1091 (57.3)
Women		734 (40)		812 (42.7)
Hypertension (%)	9 (47.4)	820 (45)	1292 (64.8)	
Diabetes mellitus (%)	8 (42.1)	568 (31)	830 (41.6)	569 (29.9)
Chronic kidney disease (%)		339 (18)		287 (15.1)
ACEI/ARB treatment history (%)	4(21.1)		655 (32.9)	195 (10.2)
Coronary artery disease (%)			289 (14.5)	
Congestive heart failure (%)		244 (13)	208 (10.4)	87 (4.6)
Liver disease (%)		99 (5)	42 (2.1)	
Peripheral vascular disease (%)		194(11)	61 (3.1)	
Cancer (%)			133 (6.7)	38 (2.0)
Obesity (%)			739 (37.1)	

## Data Availability

The data used in this article are sourced from materials mentioned in the References section.

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
