# Peer review of "Acute Kidney Injury in COVID-19"

_ijms, 2021, doi:10.3390/ijms22158081_

Round 1

Reviewer 1 Report

There are already several reviews and a consensus report addressing the issue of COVID19 and the kidney, especially acute kidney injury. In the present submission there are no major inaccuracies but several omissions pertaining to subtopics and literature. Minor modification of the review flow and correction of tables are also suggested, for the sake of clarity.

Specific points:

-Acute kidney injury and severity stages (KDIGO) should be defined at the onset of the review rather than later in its middle part on page 6 (presently AKI stages are discussed in table 1 but defined later in text).

1-Analysis of the literature concerning kidney infection by SARS-Cov2 should be refined. Reference 20 is given as the main reference for viral infection, but this study has been challenged, including in the journal where it was published, and is considered as not documenting true viral particles in the kidney. However, later studies based on different approaches support infection. The authors may consider discussing these more relevant studies, like Braun et al Lancet 2020; 396: 597-8, Puelle et al New England Journal of Medicine 2020; 383:590-92. They could find other references and topic discussions in Vijayan and Humphreys Nature Reviews Nephrology 2020; 16:703–704.

Incidentally, studies in kidney organoids also supported infectivity: Monteil et al Cell. 2020 May 14;181(4):905-913.e7.

Concerning clinical handling of the disease the following paper is worth considering and discussing: Nadim, M.K. al. COVID-19-associated acute kidney injury: consensus report of the 25th Acute Disease Quality Initiative (ADQI) Workgroup. Nat Rev Nephrol 16, 747–764 (2020).

2-Topic of ACE inhibitors, angiotensin II AT1R blockers (sartans) and putative risk of AKI in COVID19 can be briefly addressed just to state that this risk, based on theoretical considerations, has been dismissed by all adequately powered observational studies. These drugs should not be discontinued in uncomplicated COVID19. However, if AKI occurs the treatments are usually discontinued. Several references as well as recommendations from international societies (Hypertension, Cardiology, Nephrology) support this position.  

3-There are several issues with the labelling of the tables:

-Table 1: see concern about definition of AKI stages above.

-Table 2: division between electronic and light microscopy is of little cognitive or practical interest except for detecting viral particles, but with the reservation discussed above. It may be more appropriate to present morphological data according to nephron segment involved and/or putative pathophysiological mechanisms.

-Table 3 giving p values for creatinine and BUN (presumably compared to normal values) is meaningless in the context of this table. Just give values.

-Table 4 as ACEI/ARB are presented here, see point 2 above.

-Table 5 It is not clear to what the numbers of patients presented under the different items refer, especially when dual figures are shown in the same box. Consider clarifying and reporting total number of patients included and cases (AKI) and controls, when relevant.

-4 Conclusion: issue of middle-term prognosis of AKI, when not lethal, is worth addressing here, as we are now more than one year after the onset of the pandemy. Data are beginning to become available. See for example: Nugent J et al. Assessment of acute kidney injury and longitudinal kidney function after hospital discharge among patients with and without COVID-19. JAMA Network Open. Published online March 10, 2021. doi:10.1001/jamanetworkopen. 2021.1095.

Author Response

There are already several reviews and a consensus report addressing the issue of COVID19 and the kidney, especially acute kidney injury. In the present submission there are no major inaccuracies but several omissions pertaining to subtopics and literature. Minor modification of the review flow and correction of tables are also suggested, for the sake of clarity.

Specific points:

-Acute kidney injury and severity stages (KDIGO) should be defined at the onset of the review rather than later in its middle part on page 6 (presently AKI stages are discussed in table 1 but defined later in text).  

  • We moved the KDIGO criteria to paragraph 1. Introduction and referred to it later in the text. 

1-Analysis of the literature concerning kidney infection by SARS-Cov2 should be refined. Reference 20 is given as the main reference for viral infection, but this study has been challenged, including in the journal where it was published, and is considered as not documenting true viral particles in the kidney. However, later studies based on different approaches support infection. The authors may consider discussing these more relevant studies, like Braun et al Lancet 2020; 396: 597-8, Puelle et al New England Journal of Medicine 2020; 383:590-92. They could find other references and topic discussions in Vijayan and Humphreys Nature Reviews Nephrology 2020; 16:703–704. 

  • We changed reference 20 to Braun et al. 
  • We removed the sentence in paragraph 5 “Electron microscopy revealed corona-like virus-like cells in the proximal part of the renal tubules and in podocytes.”, because it refers to previous reference 20, which we have to change.

Concerning clinical handling of the disease the following paper is worth considering and discussing: Nadim, M.K. al. COVID-19-associated acute kidney injury: consensus report of the 25th Acute Disease Quality Initiative (ADQI) Workgroup. Nat Rev Nephrol 16, 747–764 (2020). 

  • We added a part “Clinical Handling” according to the article in above.

2-Topic of ACE inhibitors, angiotensin II AT1R blockers (sartans) and putative risk of AKI in COVID19 can be briefly addressed just to state that this risk, based on theoretical considerations, has been dismissed by all adequately powered observational studies. These drugs should not be discontinued in uncomplicated COVID19. However, if AKI occurs the treatments are usually discontinued. Several references as well as recommendations from international societies (Hypertension, Cardiology, Nephrology) support this position.  

  • We addressed the issue of ACEI/ARB treatment in COVID-19 in paragraph 7. Risk Factors of AKI development.

3-There are several issues with the labelling of the tables:

-Table 1: see concern about definition of AKI stages above.

-Table 2: division between electronic and light microscopy is of little cognitive or practical interest except for detecting viral particles, but with the reservation discussed above. It may be more appropriate to present morphological data according to nephron segment involved and/or putative pathophysiological mechanisms.

         -  We changed table 2. according to both reviews.

-Table 3 giving p values for creatinine and BUN (presumably compared to normal values) is meaningless in the context of this table. Just give values.

-Table 4 as ACEI/ARB are presented here, see point 2 above.

-Table 5 It is not clear to what the numbers of patients presented under the different items refer, especially when dual figures are shown in the same box. Consider clarifying and reporting total number of patients included and cases (AKI) and controls, when relevant.

  • All issues have been taken into account. We also added charts to graphically represent how AKI contributed to ICU admissions, mechanical ventilation and in-hospital death. In chart number 3 (in-hospital death) we gave up the division into AKI 1, AKI 2, AKI 3 due to insufficient data.

-4 Conclusion: issue of middle-term prognosis of AKI, when not lethal, is worth addressing here, as we are now more than one year after the onset of the pandemy. Data are beginning to become available. See for example: Nugent J et al. Assessment of acute kidney injury and longitudinal kidney function after hospital discharge among patients with and without COVID-19. JAMA Network Open. Published online March 10, 2021. doi:10.1001/jamanetworkopen. 2021.1095.

  • We addressed the provided data and added it to our conclusions. 

Reviewer 2 Report

GĹ‚owacka et al aim to provide an insight into the possible pathomechanisms of AKI in COVID-19, its clinical features, risk factors, impact on hospitalization and possible ways for its management via renal replacement therapy.

Overall, it is a good review, however a few concepts are presented very broadly, making it hard to understand. They need to be explained in more detail. I have a few comments that could be considered to improve the review.

  1. Mechanism of SARS-CoV-2 cellular kidney infection

How about kidney endothelial cells?

  1. Pathophysiology

4.1. Direct cellular infection

Please clarify the following sentence, I found it hard to understand: ‘Brush borders of renal proximal tubular epithelial cells, and at lower levels in glomerular and vascular endothelial cells (podocytes), are placed in kidneys which have high concentrations of ACE2. Therefore, this is the most likely mechanism of direct cellular infection.’

4.2. Cytokine storm and AKI related to ARDS

Please explain cardiomyopathy 'ground'

4.3. Kidney-lung crosstalk theory

Please expand on how damage to renal tubules causes alveolar capillary permeability?

4.4. Hypovolemia

‘The cause is the damage to the endothelium by the intrusion of viral cells, loss of fluid into the third space and hypotension’ How does intrusion of viral cells causes fluid loss and hypotension? please expand to make it clearer.

Please define: Rhabdomyolysis, metabolic acidosis, and hyperkalaemia

4.5. Collapsing Glomerulopathy

  1. Histopathology

‘These changes in COVID-19 patients confirm direct viral infection into cells as a mechanism of AKI.’ I think that conclusion should come at the end of the paragraph otherwise that conclusion could not be made as there's no evidence presented to justify that statement.

Optional: In table 2, perhaps adding another column to specify the type of kidney tissues e.g. Glomerular endothelial cells, podocytes or proximal tubular cells would make it helpful

  1. Risk factors of AKI development

Table 4

% should be within parentheses.

  1. Impact of AKI development on hospitalization and mortality rate

Table 5

Optional: I wonder whether this can be graphically represented, easier for the readers to instantly see the difference/impact of AKI on these

Author Response

GĹ‚owacka et al aim to provide an insight into the possible pathomechanisms of AKI in COVID-19, its clinical features, risk factors, impact on hospitalization and possible ways for its management via renal replacement therapy.

Overall, it is a good review, however a few concepts are presented very broadly, making it hard to understand. They need to be explained in more detail. I have a few comments that could be considered to improve the review.

Mechanism of SARS-CoV-2 cellular kidney infection

How about kidney endothelial cells?

  • We did not find any data on endothelial cells' involvement in the SARS-CoV-2 infection process. 

Pathophysiology

4.1. Direct cellular infection

Please clarify the following sentence, I found it hard to understand: ‘Brush borders of renal proximal tubular epithelial cells, and at lower levels in glomerular and vascular endothelial cells (podocytes), are placed in kidneys which have high concentrations of ACE2. Therefore, this is the most likely mechanism of direct cellular infection.’

  • We clarified the revised sentence: The highest concentration of ACE2 in kidneys was proven to be located in proximal tubular epithelial cells and podocytes [15, 17]. Therefore, the direct infection of kidney cells by SARS-CoV-2 virus is the most likely mechanism for the development of acute kidney injury

4.2. Cytokine storm and AKI related to ARDS

Please explain cardiomyopathy 'ground'

  • We added an explanation to the cardiomyopathy ground: Cardiomyopathy can cause stasis in the renal veins, resulting in renal hypotension and hypoperfusion, leading to a reduction in the glomerular filtration rate.

4.3. Kidney-lung crosstalk theory

Please expand on how damage to renal tubules causes alveolar capillary permeability?

  • We changed the title of the paragraph and explained how lung injury affects kidneys: lThe occurrence of kidney dysfunctions in COVID-19 patients might be explained by the kidney-lung crosstalk theory. This is due to the increased concentration of cytokines in the blood, the release of which is promoted by lung injury. Elevated levels of cytokines, especially IL-6, cause an increase in alveolar capillary permeability and pulmonary hemorrhage, and may even lead to distant-end organ dysfunction as damage to vascular endothelium in the kidneys. As a consequence, it leads to secondary hypoxia of the kidney and further damage to its structures. In patients who did not have chronic kidney disease or AKI the research has shown that, most of them manifested ARDS and / or AKI after developing pneumonia which also testifies to the lung-kidney crosstalk [28]. 

4.4. Hypovolemia

‘The cause is the damage to the endothelium by the intrusion of viral cells, loss of fluid into the third space and hypotension’ How does intrusion of viral cells causes fluid loss and hypotension? please expand to make it clearer.

  • We changed this part and explained the mechanism.

Please define: Rhabdomyolysis, metabolic acidosis, and hyperkalaemia 

  • We added definitions in parentheses.

4.5. Collapsing Glomerulopathy

Histopathology

‘These changes in COVID-19 patients confirm direct viral infection into cells as a mechanism of AKI.’ I think that conclusion should come at the end of the paragraph otherwise that conclusion could not be made as there's no evidence presented to justify that statement.

  • We added a conclusion at the end of the paragraph.

Optional: In table 2, perhaps adding another column to specify the type of kidney tissues e.g. Glomerular endothelial cells, podocytes or proximal tubular cells would make it helpful

  • We changed table 2. according to both reviews.

Risk factors of AKI development

Table 4

% should be within parentheses.

  • We added parentheses.

Impact of AKI development on hospitalization and mortality rate

Table 5

Optional: I wonder whether this can be graphically represented, easier for the readers to instantly see the difference/impact of AKI on these.

  • We made charts that graphically represent how AKI contributed to ICU admissions, mechanical ventilation and in-hospital death. In chart number 3 (in-hospital death) we gave up the division into AKI 1, AKI 2, AKI 3 due to insufficient data.

Round 2

Reviewer 1 Report

The authors have been responsive to the review and the paper has been improved, both for content and presentation, at revision.